# Firm Size and Age mediating the Firm Survival-Hedging Effect: Hayes' 3-Way Parallel Approach

**Henry Okwo**, **Charity Ezenwakwelu**, **Anthony Igwe *** and **Benedict Imhanrenialena**

Department of Management, University of Nigeria, Enugu Campus, Enugu 400001, Nigeria;
henry.okwo.pg81372@unn.edu.ng (H.O.); charity.ezenwakwelu@unn.edu.ng (C.E.);
ogbemudiabenedict@yahoo.com (B.I.)
* Correspondence: anthonyigwe121@yahoo.com

**Abstract:** A James Gaskin Excel Macro Analysis is performed to determine the reliability of our scales, and a 3-way parallel mediation using the Andrew Hayes' PROCESS model is applied to test the formulated hypotheses. Results show that hedging has a direct effect on firms' survival; firms' size and age individually do not strongly influence these effects, but a combination of the two does. We, therefore, concluded that while the hedging-survival effect exists on all forms of hedging, the practice of hedging is consequential for firms on the premise of their ages and numbers of employees.

**Keywords:** firm size; hedging; firm age; firm survival; mediation; Hayes

---

## 1. Introduction

Firm size and age are major factors associated with firm survival [1–5]. While there is an assumption that small firms grow more rapidly [1], one could still argue that they only do so because they also exit the marketplace more quickly than large firms [6]. With respect to the argument about the age and size of firms, Jovanovic [7], in a seminal paper, asserts that it is as firms engage in learning the industrial terrain wherein they exist that they inadvertently know how efficient, or otherwise, they are. He further states "Efficient firms grow and survive; inefficient firms decline and fail" (Jovanovic [7], p. 2).

It is expected that efficient firms actually should strive for longer periods, while less efficient firms will shortly exit the market. This idea as encapsulated by Jovanovic [7] in his "Theory of Noisy Selection" (TNS), which assumes that all firms and employees start on an equal pedestal, irrespective of size, but while their survival is not dependent on their size, it is rather dependent on the level of adjustment efficiency attained with time. The premise of adjustment is a prime assumption of another theory, Adjustment Cost Theory (ACT). This theory supposes that the relationship between inefficiency and extinction is nonlinear, but requires inefficient firms to constantly adjust their cost structure, mostly in terms of employees' formation, numbers and output [8–11]. TNS also assumes that there are unobservable gyrations (noise) which a firm has to grapple with, and it is a given firm's level of information on how much 'noise' it has been able to manage that keeps it in the industry.

Another theory that supports the postulation of survival assumption of TNS and ACT is the Schumpeterian Growth Theory (SGT), or the theory of creative destruction. This theory supports the premise of efficiency on the basis of innovation as being a precursor to whether a firm survives competition, and grows or dies [12,13]. Like ACT, SGT implies that questions of employee size (structure), age, and the profit range of a firm are predicted by what the firm either does or fails to do. Goedhuys [14], Pastor & Veronesi [15], and Arkolakis, Papageorgiou & Timoshenko [16], in support of these theories, have linked another theory to the issue of firm growth (survival): Active Learning

Theory (ALT). ALT holds that as long as a firm is an active market player, whether adjusting its cost or being innovative, constantly responding to the market dynamics at play within the industry in order to survive, that firm would grow and expand [17]. In contrast to ALT is the Passive Learning Theory (PLT), which rather submits that all a firm needs to survive is to be efficient at its internal processes [18]. This theory assumes that as long as a firm has been able to grapple with enough 'noise', such a firm can count itself as being efficient, and would remain longer in the industry [19,20]. This is synonymous with TNS. We base this study on these theoretical assumptions, and while we admit that the theories are much more complex and emphasize mathematical models, we intend to make our contribution to the ecology and evolution of firms a much more simplistic and policy-friendly study to the Nigerian business environment.

Businesses in Nigeria have long-term survival issues [21,22], and the survival of corporate entities is directly related to national sustainability [23,24]. While there could be various factors mitigating against the long-term survival of these businesses, we intend to, in a scientific, systematic, but simplistic manner, investigate how one important issue predicts the survival of these businesses. This issue is *managing foreign exchange rate risk*. Firms in Nigeria operate in a global business environment, and although they may be unaware of their roles of having to serve the largest market in Africa, they are obviously being battered by their inability to handle fluctuations in the global foreign exchange market [25–28]. Like TNS, most of these firms may be passive, while simply relying on their age and size, but 'noise' from the foreign exchange markets may lead to the outright extinction of most of these firms [29–31]. The ability, therefore, of firms to handle the gyrations and constant fluctuations in the financial market is known as 'hedging' [32–37]. The hedging behavior of firms is meant to increase their age and size, leading to improved survivability of these firms. The research questions of this study are: do firms' hedging behaviors improve their survivability? What mediating role would a firms' age and size have on the hedging-survival effect? We consider this in consonance with the Hayes approach, and not that of Baron and Kenny, for reasons discussed in the body of this paper.

The subsequent parts of this paper contain a review of the relevant empirical and conceptual literature upon which the hypotheses are anchored, and a section on the methodology which serves to define the variables, methodological design, data collection technique and the method of statistical test employed. This is followed by the presentation of the results of a hypotheses test, discussions of findings, recommendations and policy implications, contribution to knowledge, and conclusion and avenue for further studies.

## 2. Literature Review

### 2.1. Managing Foreign Exchange Rate Risks and Firm Survival

Soenen, [38], Kelley [39], Hagelin [40], Noreiko & Solga, [41], Hill, [42], Hekman [43] and He & Ng [44] all recognize that hedging occurs when managing the exposure of short-term transactions (transaction exposures), managing records affected by foreign exchange fluctuations (translation or accounting exposures) and managing long-term implications of fluctuations in the foreign exchange market (economic exposures), although there have been attempts to decompose hedging differently [45,46]. Firms may hedge consciously in consonance with the ALT, or unconsciously, as assumed by the PLT, but we argue that absolute disregard of hedging may not augur well for these manufacturing firms, irrespective of their size [47,48]. The presumption that a firm is too small to hedge is preposterous, because the commodities which that firm produces are sold in the same market where other imported commodities are sold, and for bigger firms that may be export-oriented, the commodities which they export are also being sold in foreign markets with much more hedging-conscious firms to the detriment of the non-hedgers, impeding their life-cycle [49–51].

The theoretical evidence of the life-cycle which characterizes the lifespan of every organization has kept the management of organizations on their toes to avoid decline or possible death [52]. Manufacturing firms, just like every other business organization, seek to remain in business for a

very long time, even to perpetuity [20,53]. To achieve this, firms explore the means to overcome several challenges which the business environment constantly poses to their operations [54,55]. The scarcity of raw materials and costs of financing production processes amongst others are challenges to manufacturing firms. These challenges, coupled with globalization, have prompted firms to compete at a global level [56,57]. As such, import-based firms, export-based firms or firms in both orientations constantly seek to minimize costs at all levels of their operational processes, so as to remain competitive [51,58]. Costs associated with transactions, translations and economic exposure constitute risks to firms. This demands constant proper management, as it, to a large extent, affects the survival of a firm.

Firm survival, as implied in SGT, is the "conscious destruction" of strategies that have not served the firm's ability to withstand competition both within its immediate market and the global market environment [59,60]. The subtle adoption of better strategies which lead to the creation and development of new ideas which are consequential to meeting new demands implies, therefore, that a firm has high survival ability [48,54,61,62]. Survival is showing resilience amidst dynamic and turbulent business environments to meet corporate needs and avoid being edged out. This is usually anchored on the proper management of risk exposure by a firm [37,63–65]. In line with organizational learning theories as earlier discussed, active firms acquire financial information through deliberate forecasting and try to hedge against fluctuations in the foreign exchange market, thereby withstanding shocks and noise better than their competitors. This should inevitably keep the firm on a better survival path than others in the industry [56,63,64,66].

On this premise, the hypothesis below is formulated, conveying the a priori expectations of the authors:

**Hypothesis 1:** Transaction (TRC), Translation (TRSL) and Economic (ECO) exposure management, as learning and innovative strategies are predictors of firm survival (fSurv.).

*2.2. The Mediating Role of Firm size and Age*

One important thing to note from the above is that although hedging could be associated with different firms' outcomes [33,34,36,43,67–70], complexities are inherent in such associations. For one, firms exist and operate in dynamic/hostile environments [71–73], and these environments play both controlling and intervening roles in these associations. We considered how the firms' sizes affect their ability to survive at one extreme, while looking at how it is affected by the firms' risk management strategies at the other. The rationale behind this consideration is based on the contributions of Hannan, Carroll, Dobrev & Han, [74], Albuquerque & Hopenhayn, [75], Gallo & Christesen, [76] and Esteve-Perez, Pieri & Rodriguez, [5], that a firm's size and age are fundamental variables that affect organizational outcomes; therefore, little changes may have a significant impact on these outcomes in an empirical study.

A firm's size is defined by both financial and nonfinancial measures, such as profit and employee numbers respectively [19,77–79]. On the other hand, the age of a firm as a mediator is operationalized as how long a firm has been in active business engagement [30,51,80]. In this regard, most studies have conceptualized the ages of firms by a categorization into either young or old firms [4,5].

For this study, we adopted three proxies as mediators: firm size, which includes both employees and profit sizes, and firm age. Employee numbers as a proxy of firm size is important to the firm, because it defines the scope of a firm as being micro, small, medium or large [81]. Profit as the second firm size proxy is the total return to the implicit (entrepreneurial) inputs, and it is this magnitude which is maximized [82,83]. As a second mediator, the firms' age is the length of the period a firm is engaged in actual business operations [80]. This could be defined as operations bordering on fiscal responsibilities and administrative engagements [51,79]. These proposed mediators are also expected to intervene between the predictor and outcome variables that basically should have a direct

effect. This is where our study diverged from previous studies on firm size and age as controlled variables [5,36,84–86]. This means that the size of a firm in terms of employee numbers and profit levels, coupled with the age of the firm, would mediate with various exposure managements to either reduce or increase the probability of that firm's survival. Therefore, we propose the following hypotheses:

**Hypothesis 2:** Firm Age, Employee, and Profit Size mediate the effect of TRC, TRSL, ECO on fSurv.

**Hypothesis 3:** There is a total effect of Firm Age, Employee and Profit Size combined with TRC, TRSL, and ECO on fSurv.

## 3. Materials and Methods

We developed questionnaire instruments for this study. While we designed question items anew for the hedging constructs, we also focused on the items and measures of firm survival. Firm survival as an outcome variable of this study was selected for its prime place in the ecology and development of industry [1,58,62,87–89]. To measure this variable, we adopted a highly scientific, but simplistic approach. While other studies have focused on survival from various econometric and parametric quantifications [41,90], we instead adopted a much more easy-to-understand approach, which makes for easy policy direction and use [91]. The measures for both scales were developed by the authors of this paper. This was done in consonance with the existing literature on the concepts of hedging [36,38,43,65,67,69,92–98]. From the cited studies, scales were developed on managing TRC, being managing short-term business transaction exposures experienced by firms; TRSL, being management of accounting and reporting errors based such foreign exchange exposures; ECO-management of long-term exposures, these being the independent variables of the study and firm survival (dependent variable) [1,53,58,99], emphasizing the firm's drive for development, its intention to quit and third party perception, as reported in the Exploratory Factor Analysis (EFA) presentation. The mediating variables of the study are the size of the firm (employees and profit) and the age of the firm. These items were developed being premised on theoretical and empirical considerations while paying keen interest to studies that have also adopted a survey approach.

The survey was conducted in collaboration with Godfrey Okoye University, Enugu State, Nigeria; Researchers for Sustainable Societies (RSS); Researchers for Contemporary Issues in the Business Circle (RCIBC) of the Faculty of Business Administration of University of Nigeria and University of Nigeria Business School. The survey was carried out in South Eastern Nigeria, and benchmarked the Ngozi Okonjo-Iweala (NOI) Polls, which sets the highest standard for opinion-based research in Nigeria. In 2017, NOI released its report on the Nigerian manufacturing sector, titled "Manufacturing Sector: Operating amidst economic recession and unsteady foreign exchange rates, 2017". In that report, which was produced in collaboration with the Centre for the Study of Economies of Africa (CSEA), 75% of Nigerian Manufacturing firms opined that foreign exchange rate "disparity" was their biggest challenge, and that it affected them negatively [28]. Our instrument being aligned with the NOI polls supports the fact that opinion-based research can be used to address issues such as foreign exchange rate fluctuations amongst firms. We also prepared questionnaires which were informed by the NOI consideration. Of the 496 distributed questionnaires, 351 (70.8%) were properly filled out and returned. The respondents were firm owners, managers, directors and whoever is at the decision making level of the firm. This survey was conducted between December 2017 and June 2018. All questionnaires were self-administered by members of the research groups mentioned, all being at least doctoral students and of the Management discipline (See "Results" section for data diagnostics).

We adopted the use of a mediated regression analysis which establishes a causal effect between the predictive variable to the mediators and then the outcome, where these effects must be anchored on theory. This statistical concept is made clear by the contributions of Andrew F. Hayes [100]. We hypothesized that the effect of efficient and innovative practices as hedging should predict the

age, employee size, and profit sizes, which are mediators, and that these mediators then predict firm survival. In light of the multiple flaws and critiques of the Baron and Kenny Mediation Approach [101–105], we used Hayes' PROCESS plugin on SPSS. The Hayes Test has become a very useful tool for testing the mediation hypotheses. This tool suggests the use of "model 4", which supports the conceptual models of this study, having to test three mediators simultaneously. This method requires the tests of three effects, i.e., direct effects, specific indirect effects, and total effects. Furthermore, we use the bias-corrected bootstrapping method to establish the statistical nature of the indirect effects (Figure 1).

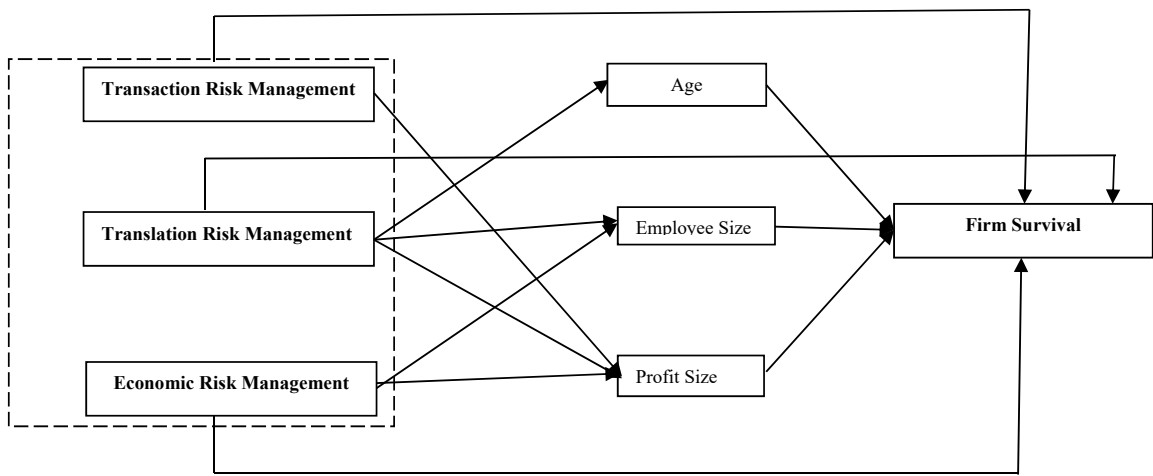

**Figure 1.** Conceptual Model.

## 4. Results

From Table 1, it was revealed that the age of the firms has a positive significant relationship with firms' survival (r = 0.231; $p < 0.01$). This result implies that the number of years the firms have been in operation has a direct relationship with the survival of the firm. This finding is consistent with those of previous studies [66,89]. There is also a positive significant relationship (r = 0.123; $p < 0.05$) between employee numbers and firms' survival, meaning that as the level of employees within the firm increases, the firm's chances of survival will increase. When correlated with the profit levels, firms' survival revealed that there is a positive significant relationship; (r = 0.265, $p < 0.01$). There is a positive significant relationship between TRC and firms' survival as well (r = 0.154, $p < 0.01$), meaning that there is a clear influence of transaction exposure management on the survival of the firm, same with the positive and significant relationships existing between TRSL, ECO and firms' survival respectively. (r = 0.265; $p < 0.01$ & r = 0.180; $p < 0.01$).

**Table 1.** Descriptive statistics of study variables.

|  | AVE | C.R | α | Mean | S.D | 1 | 2 | 3 | 4 | 5 | 6 | 7 |
|---|---|---|---|---|---|---|---|---|---|---|---|---|
| Age (1) | - | - | - | 3.5299 | 0.68751 | - | | | | | | |
| Emply (2) | - | - | - | 2.0313 | 0.42310 | −0.097 | - | | | | | |
| Profit (3) | - | - | - | 2.7493 | 0.65014 | 0.151 ** | 0.184 ** | - | | | | |
| TRC (4) | 0.725 | 0.911 | 0.868 | 16.2877 | 2.35550 | 0.142 ** | 0.037 | 0.066 | 0.851 | | | |
| TRSL (5) | 0.590 | 0.847 | 0.863 | 15.5014 | 3.16126 | 0.143 ** | 0.032 | −0.095 | −0.123 * | 0.768 | | |
| ECO (6) | 0.645 | 0.874 | 0.877 | 16.3875 | 2.12153 | 0.176 ** | 0.171 ** | 0.029 | 0.193 ** | −0.023 | 0.803 | |
| fSurv (7) | 0.778 | 0.912 | 0.903 | 11.7493 | 2.57124 | 0.231 ** | 0.123 * | 0.265 ** | 0.154 ** | 0.265 ** | 0.180 ** | 0.882 |

The signs * implies a significance at $p < 0.05$; ** at $p < 0.01$. AVE: Average Variance Extracted; C.R: Composite Reliability; α: Cronbach's Alpha.

This study adopted an Exploratory Factor Analysis (EFA) to establish the validity of the study items. This is in contrast to the use of the Confirmatory Factor Analysis (CFA), as adopted by many studies. The reason behind the use of the EFA as opposed to the CFA in this study is because the scales

used for this study were not adapted from past research; rather, the authors themselves developed the scales. Moreover, the EFA is considered to be appropriate in the early stages of scale development because it shows how well items are loaded on the non-hypothesized factors [106]. Also, to ensure that the presence of 'noise' is infinitesimal, we conducted a Common Latent Factor test. This showed a Harman's Single Factor that explained the variance of 22%, and the common heuristic factor of 31%, explaining a squared variance of 9%.

The fifteen (15) question items were associated with four (4) different factors, i.e., Transaction exposure management coded as TRC; Economic exposure management coded as ECO; translation exposure management coded as TRSL and Firms' survival coded as fSurv. Of the 15 items, 4 of them loaded under TRC thus; TRC2 = 0.935, TRC3 = 0.923, TRC1 = 0.912 and TRC4 = 0.680. Four items also loaded under ECO thus: ECO3 = 0.930, ECO2 = 0.901, ECO4 = 0.786 and ECO1 = 0.761. Four items also loaded under TRSL as well: TRSL2 = 0.910, TRSL1 = 0.835, TRSL3 = 0.810 and TRSL4 = 0.772. Finally, 3 items were loaded as one under fSurv, thus: fSurv2 = 0.922, fSurv3 = 0.903 and fSurv1 = 0.870. According to Izquierdo, Olea & Abad, (2014) [107] factors that load from 0.3 or 0.4 above are considered high, and therefore, fit for interpretation. We thus pegged the acceptance levels for our items at 0.5; this was to ensure statistical adequacy [108] (Table 2).

**Table 2.** Exploratory Factor Loading.

| Variables | Item | Mean | Std. Deviation | Loading |
|---|---|---|---|---|
| Transaction Exposure Management | My Bank/Bureau De Change helps me out during FX fluctuations in transactions | 4.1254 | 0.67291 | 0.935 |
| | At times we delay payments because of FX fluctuations | 4.0912 | 0.57838 | 0.923 |
| | We have a system to handle FX fluctuation during transactions | 4.1168 | 0.57622 | 0.912 |
| | We usually try to forecast FX trends and | 3.9544 | 0.90281 | 0.680 |
| Translation Exposure Management | We usually face disappointments whenever there is an FX fluctuation | 3.8718 | 0.99316 | 0.910 |
| | We project correctly so our records are mostly correct irrespective of FX fluctuations | 3.8519 | 0.98023 | 0.835 |
| | We've once noticed that our books did not balance as planned because of an FX fluctuation | 3.8376 | 0.99677 | 0.810 |
| | In other to stick to our financial plan irrespective of FX fluctuations, we enter agreements with our clients. | 3.9402 | 0.76671 | 0.772 |
| Economic Exposure Management | We run other businesses because of the possibility of an FX crisis | 4.0769 | 0.59743 | 0.930 |
| | Our firm has plans to likely move away from this line of business because of future FX fluctuations | 4.1624 | 0.65409 | 0.901 |
| | Whenever a business refuses to be flexible with FX deals, we cut such business ties/relationship | 4.1168 | 0.62840 | 0.786 |
| | Projections for FX fluctuations are not effectives and we do not need them here * | 4.0313 | 0.60156 | 0.761 |
| Firms' Survival | The management of this firm has never relaxed in taking drastic steps for the good of the business | 3.9630 | 0.81156 | 0.922 |
| | We intend to change from this business line and location soon * | 3.8519 | 0.94157 | 0.903 |
| | Other businesses see us as resilient | 3.9345 | 1.04128 | 0.870 |

* Reverse-coded question items.

In order to confirm the internal consistency for the items, we ran the James Gaskin Excel Macro. The analysis revealed that TRC, which consisted of 4 items, had an Average Variance Extraction (AVE) of 0.725; a Composite Reliability (CR) of 0.911 and a Cronbach's Alpha ($\alpha$) of 0.868. The TRSL, which consisted of 4 items as well had an AVE of 0.590, CR of 0.847 and $\alpha$ of 0.863. In the same vein, the ECO scale which consisted of 4 items revealed an AVE of 0.645, CR of 0.874 and $\alpha$ of 0.877. Finally, the fSurv scale consisted of 3 items with an AVE of 0.778, CR of 0.912 and $\alpha$ of 0.903.

*4.1. Hypotheses Testing*

Figure 2 represents the statistical description of the earlier stated hypotheses of the study, and also has the Statistical Equations for the hypothesized effect.

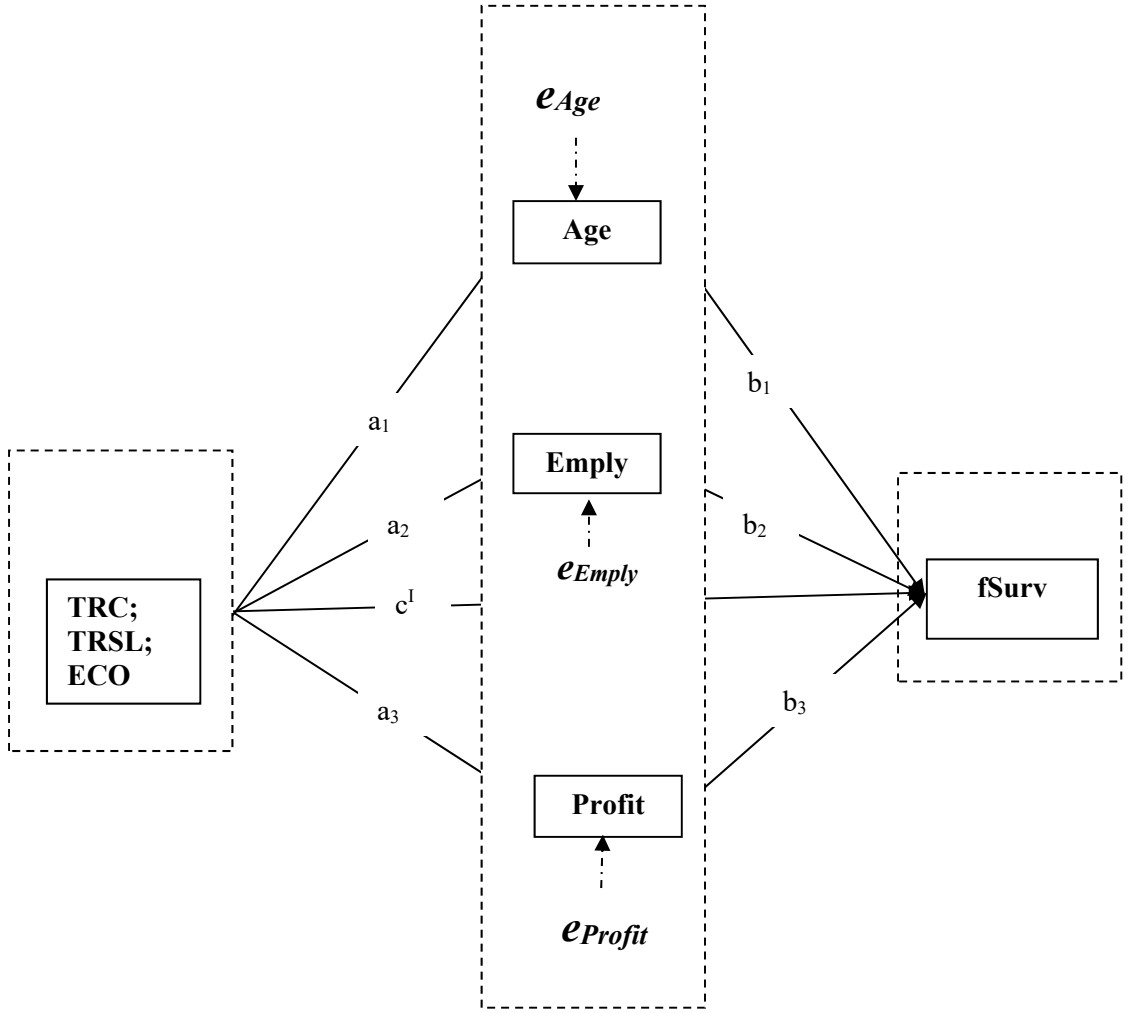

**Figure 2.** Statistical model of the hypothesized effects.

### 4.1.1. Direct Effects

Firm Age ($a_1$) = $i_{Age}$ + $a_1$ (TRC;TRSL;ECO) + $e_{Age}$
Employee Size ($a_2$) = $i_{Emply}$ + $a_2$ (TRC;TRSL;ECO) + $e_{Emply}$
Profit Size= $i_{Profit}$ + $a_3$ (TRC;TRSL;ECO) + $e_{Profit}$
Firm Survival= $i_{Surv}$ + $C^I$ (TRC;TRSL;ECO) + $e_{Surv}$

### 4.1.2. Total Effects Direct Effects

Total ($C_{(TRC;TRSL;ECO)}$) = $C^I_{(TRC;TRSL;ECO)}$ + $a_1b_1$ + $a_2b_2$ + $a_3b_3$, where $a_1b_1$, $a_2b_2$ and $a_3b_3$ are specific indirect effects (Firm age, Employee size & profit size respectively) through the predictor variables (TRC;TRSL;ECO) to the output (Surv.) (Table 3).

**Table 3.** Results on hypotheses tests.

| Predictors | | M1 (Agefirm) Coeff. | SE | p | | | M2 (Emplsize) Coeff. | SE | p | | | M3 (Profit) Coeff. | SE | p | | | Firms' Survival Coeff. | SE | p |
|---|---|---|---|---|---|---|---|---|---|---|---|---|---|---|---|---|---|---|---|
| | | | | | | | | | | | | | | | | | | | |
| **Section i** | | | | | | | | | | | | | | | | | | | |
| TrC | $a_{11}$ | 0.0414 | 0.0155 | 0.0077 | $a_{12}$ | | 0.0066 | 0.0096 | 0.4920 | $a_{13}$ | | 0.0182 | 0.0147 | 0.2181 | $c^1_1$ | | 0.1196 | 0.0554 | 0.0317 |
| Agefirm | | - | | | | | | | | | | | | | $b_{11}$ | | 0.7208 | 0.1932 | 0.0002 |
| Emplsize | | - | | | | | | | | | | | | | $b_{12}$ | | 0.5991 | 0.3131 | 0.0565 |
| Profit | | - | | | | | | | | | | | | | $b_{13}$ | | 0.8317 | 0.2051 | 0.0001 |
| Constant | $i_{Age}$ | 2.855 | 0.2545 | <0.001 | $i_{Emply}$ | | 1.9237 | 0.1581 | <0.001 | $i_{Profit}$ | | 2.4530 | 0.2426 | <0.001 | $i_{Surv}$ | | 3.7535 | 1.2677 | 0.0033 |
| | | $R^2 = 0.0202$ | | | | | $R^2 = 0.0014$ | | | | | $R^2 = 0.0043$ | | | | | $R^2 = 0.1291$ | | |
| | | $F(1349) = 7.1807,$ | | | | | $F(1349) = 0.4731,$ | | | | | $F(1349) = 1.5223,$ | | | | | $F(4346) = 12.8274,$ | | |
| | | $p = 0.0077$ | | | | | $p = 0.4920$ | | | | | $p = 0.2181$ | | | | | $p < 0.0001$ | | |
| **Section ii** | | | | | | | | | | | | | | | | | | | |
| TRSL | $a_{21}$ | 0.0311 | 0.01115 | 0.0073 | $a_{22}$ | | −0.0127 | 0.0071 | 0.0752 | $a_{23}$ | | 0.0066 | 0.0110 | 0.5483 | $c^1_2$ | | 0.1991 | 0.0403 | 0.0000 |
| Agefirm | | - | | | | | | | | | | | | | $b_{21}$ | | 0.6593 | 0.1878 | 0.0000 |
| Emplsize | | - | | | | | | | | | | | | | $b_{22}$ | | 0.7588 | 0.3055 | 0.0005 |
| Profit | | - | | | | | | | | | | | | | $b_{23}$ | | 0.8194 | 0.1995 | 0.0135 |
| Constant | $i_{Age}$ | 3.0480 | 0.1823 | <0.001 | $i_{Emply}$ | | 2.2286 | 0.1128 | <0.001 | $i_{Profit}$ | | 2.6468 | 0.1741 | <0.001 | $i_{Surv}$ | | 2.5411 | 1.1429 | 0.0268 |
| | | $R^2 = 0.0204$ | | | | | $R^2 = 0.0090$ | | | | | $R^2 = 0.0010$ | | | | | $R^2 = 0.1757$ | | |
| | | $F(1349) = 7.2797,$ | | | | | $F(1349) = 3.1846,$ | | | | | $F(1349) = 0.3611,$ | | | | | $F(4346) = 18.4315,$ | | |
| | | $p = 0.0073$ | | | | | $p = 0.0752$ | | | | | $p = 0.5483$ | | | | | $p < 0.001$ | | |
| **Section iii** | | | | | | | | | | | | | | | | | | | |
| ECO | $a_{31}$ | 0.0571 | 0.0171 | 0.0009 | $a_{32}$ | | 0.0341 | 0.0105 | 0.0013 | $a_{33}$ | | 0.0089 | 0.0164 | 0.5856 | $c^1_3$ | | 0.1556 | 0.0629 | 0.0138 |
| Agefirm | | - | | | | | | | | | | | | | $b_{31}$ | | 0.6820 | 0.1949 | 0.0005 |
| Emplsize | | - | | | | | | | | | | | | | $b_{32}$ | | 0.4742 | 0.3183 | 0.1371 |
| Profit | | - | | | | | | | | | | | | | $b_{33}$ | | 0.8666 | 0.2047 | 0.0000 |
| Constant | $i_{Age}$ | 2.5944 | 0.2822 | <0.001 | $i_{Emply}$ | | 1.4723 | 0.1738 | <0.001 | $i_{Profit}$ | | 2.6026 | 0.2709 | <0.001 | $iSurv$ | | 3.447 | 1.282 | 0.0075 |
| | | $R^2 = 0.0310$ | | | | | $R^2 = 0.0293$ | | | | | $R^2 = 0.0090$ | | | | | $R^2 = 0.1328$ | | |
| | | $F(1349) = 11.178,$ | | | | | $F(1349) = 10.5187,$ | | | | | $F(1349) = 1.5223,$ | | | | | $F(4346) = 13.24,$ | | |
| | | $p = 0.0009$ | | | | | $p = 0.0013$ | | | | | $p = 0.5856$ | | | | | $p < 0001$ | | |

TRC stands for Transaction Risk Management; TRSL stands Translation Risk Management; ECO stands for Economic Risk Management; Agefirm stands for the Age of the Firm; Emplsize stands for the size of the firm employees; Profit stands as the profit size of the firms.

Figure 3 shows that there is a direct correlation between hedging and firms' survival. As stated in the first sets of hypotheses, transaction exposure management, translation exposure management and economic exposure management affects the survival of firms. These hypotheses were supported. Specifically, Figure 3 shows that transaction exposure management has a direct effect on firms' survival ($c_1' = 0.12$; t (346) = 2.16; $p < 0.05$); translation exposure management has a direct effect on firms' survival as well ($c_2' = 0.1991$; t (346) = 4.94; $p < 0.0001$). In the same vein, the hypothesis that economic exposure management has a direct effect on firms' survival was also supported ($c_3' = 0.1556$; t (346) = 2.475; $p < 0.05$).

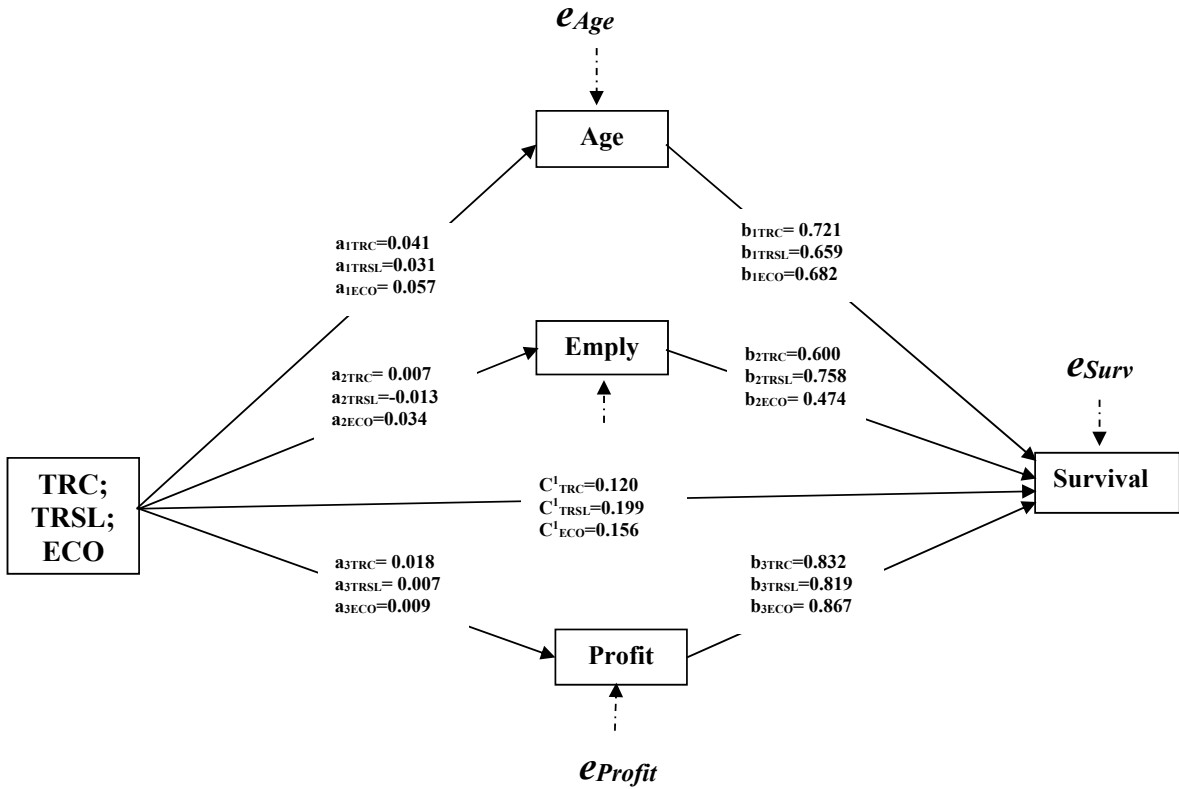

**Figure 3.** Summary of effects.

In line with the second sets of hypotheses which stated that age, number of employees and profit levels mediate the hedging- firms' survival relationship, we garner little and no support. Specifically, the hypothesis that age influences the firms' survival through transaction exposure management was not supported ($a = -0.0025 \leq a_1 \leq 0.0663$). Employees numbers and transaction exposure management-survival relationship was also not supported ($e = -0.0136 \leq e_1 \leq 0.0248$), and finally, the influence of firms' profit levels on the transaction exposure management-survival relationship ($p = -0.0150 \leq p_1 \leq 0.0442$) was also not supported.

In contrast, a not so similar result was obtained; thus, the hypothesis that translation exposure management influences the firms' survival through age was supported ($a = 0.0205$; $0.0414 \leq a_2 \leq 0.0414$). The hypothesis that employee numbers mediates on the effects of translation exposure management on firms' survival was not supported ($e = -0.0097$; $-0.0282 \leq e_2 \leq 0.0007$). The hypothesis that firms' profit levels mediate the translation exposure management-survival effect was also not supported ($p = 0.0054$; $-0.0182 \leq p_2 \leq 0.0385$).

The ECO-fSurv hypothesis showed that age mediates on the economic exposure management-survival effect ($a = 0.0389$, $0.0131 \leq a_3 \leq 0.0769$); the size of employees within a firm mediates the effects of economic exposure management on the survival of the firm ($e = 0.0162$,

$0.0002 \leq e_3 \leq 0.0354$). Profit levels do not support our proposition as a mediator on the effects of economic exposure management on firms' survival ($p = 0.028$, $-0.01665 \leq p_3 \leq 0.0457$).

The last set of results showed the *Total effects* of the various hedging strategies on the survival of firms when combined with the mediators. Specifically, transaction exposure management when combined with the sum of the mediators, revealed a positive and significant effect; ($C_1 = 0.1686$; $0.0550 \leq c \leq 0.2821$; $p < 0.05$); the total effects of translation exposure management combined with the mediators on the survival of firms is also positive and significant ($C_2 = 0.2153$; $0.1327 \leq c \leq 0.1327$; $p < 0.05$); the overall significant positive effect between economic exposure management combined with the mediators on firms' survival was confirmed as well ($C_3 = 0.2185$; $0.092 \leq c \leq 0.3439$; $p < 0.05$).

## 5. Discussions

### 5.1. Direct Effect

The study found a direct effect of all forms of hedging on the survival of the firm. The path weights or effect size showed that translation exposure management predicts firm survival better than economic and transaction exposure management. This implies that a unit change in the degree to which organizations try to mitigate the fluctuations in the foreign exchange markets will affect the survival ability of such firms. This finding is supported by several studies [2,63,65], which have shown that foreign exchange rate risk and its attendant fluctuation is a consequential issue that businesses face, especially in developing economies. Also, the SGT's assumption that firms who refuse to innovate or recreate themselves, in being dynamic, would definitely find themselves to be unable to weather the challenges posed by the macro-factors of the global economic system. In contrast, Ahmed, et al., (2013) [35] found that not all forms of risk management undertaken by a firm benefits them. They argue that hedging interest-rate risk may not lead to a positive outcome. There was also an earlier study in which it was found that selective hedging may not necessarily lead to any clear benefit [109], while others have found that it is the size of the portfolio being hedged or protected that determines whether a hedging exercise is going to yield positive rewards. In light of this, we submit that for firms in developing economies, operating in an ever-dynamic market environment, any hedging activity that shields these firms from the negative consequences of operating in the same market with multinationals who may have an abundance of information, and most likely a corporate hedging policy, would be clearly helpful in surviving the turbulence caused by the gyrations of the foreign exchange market.

### 5.2. Indirect Effects

On specific indirect effects, transaction exposure (short-term) management did not mediate with any of the study variables in predicting the survival of the firm. Translation exposure management is significantly mediated by the age of the firm in predicting survival, but is not statistically mediated by employee and profit size. On economic exposure management (long-term), firm survival is significantly predicted through both age and employee size as mediators. One glaring finding here is that profit size is not a significant mediator to any of the three (3) forms of hedging [110,111], supporting these findings. They, among others, suggest that firm age and employee numbers could be an outcome of innovative or efficient organization processes, but not necessarily profit size [112,113]. On the other hand, Esteve-Perez et al. [5] and Bellone et al. [114] are of the opinion that profitable firms are firms that have been more efficient and are more likely to survive, being that they have a lower hazard risk. While we understand the theory behind Esteve- Perez and Bellone's colleagues' findings, we also feel that profit may be a factor controlled by age, and firms would have to break even with time first, before declaring profits. On age as a mediator, it is not surprising that it plays no role with transaction exposure management, as this is a short-termed activity, while age focuses on longer-term issues which are shown by their effects as mediators for both the translation and economic exposure management. Numerous studies have linked age with both hedging and firm value, survival or

performance [30,31,66,89,115], stating that older firms survive longer, clearly placing age as a predictor of survival fit and on the outcome of efficient practices.

*5.3. Total Effect*

The findings in the total effect of the predictor variables (Transaction exposure management, Translation exposure management, and Economic exposure management) through the mediators (Firm age, Employee size and profit size) are similar to the results of the direct effect. This implies the presence of a complementing mediation. This simply implies that all direct effects correspond/complement the total effects. Also, it is noteworthy, as stated in the methodology, that this report style is in contrast to that of Baron & Kenny, where they would have labeled our result as a partial mediation. Rather we argue that mediation is mediation, and the increase in direct effects implies that the mediators are positive and significantly contribute to the survival of the study firms. None of the mediating variables reduced the impact of the predictive direct effects on the survival of the firms. As stated earlier, multiple studies support the idea that these selected variables would be outcomes of strategic decisions of efficiency and innovativeness, while also supporting the fact that these variables could lead to the improved survival of firms. In contrast, there are studies that do not feel this way, and some even feel that the effect model could be the other way round, that is, age predicting hedging, rather than hedging predicting age. In all, we submit from our findings that for firms in developing markets to remain active in the market, speculative operators and non-hedgers would have to be educated; this is the clear precursor for both age and even improved employment [116], while the survival of such firms is not an issue that age or employee size merely predicts, as profits (see Figure 3) are equally also a critical predictor of survival. So, rather than submitting to the Jovanovic's logic of passive learning and noisy selection, stating that age is the main predictor of survival, we find that profit rather is, but not in a mediation model.

## 6. Conclusions

Manufacturing firms, like every other business type, no doubt face turbulent situations in an dynamic environment. In the face of global competitiveness, manufacturing firms face foreign exchange volatility with the attendant transaction, translation and economic risks exposure which affects their operational activities. The proclivity to learn and seek dynamics of achieving firms' purposes through innovation, as supported by Schumpeterian growth theory, seemed consequential to managing these risks which are crucial for survival [13,90].

Generally, we established that all the hedging measures directly predict firm survival. Firm age, employee size, and profit size do not mediate between transaction exposure management on firm survival. However, firm age mediates between the effects of both translation and economic exposure management on firm survival, while employee size mediates between economic exposure management and firm survival. All the models had a significant total effect.

## 7. Recommendations and Policy Implications

In the face of the dynamic business environment coupled with global competitiveness evidenced on the direct effect of hedging on firm survival, we recommend that firms should be active players in understanding foreign exchange dynamics, as this would help them in managing the associated risks to enhance their survival. Based on theoretical evidence, firms that want to survive amidst global competition should be able to learn the nitty-gritty of foreign exchange dynamics and creatively do away with none-workable procedures that may expose them to the risk of extinction. Foreign exchange volatility is a global issue which affects firms, and governments should not wade against the tide of global best practices in shielding local firms on foreign exchange dynamics. Though governments have the duty to develop policies that protect emerging national firms from the impact of fluctuations inherent in foreign exchange volatilities, this should be done while benchmarking other advanced nations' working policies on foreign exchange management.

Specifically, firms should place less emphasis on profit as a basis hedging, as our findings showed that age and employ numbers mediate between hedging and survival relationship, while profit had no significant effect. The government should provide adequate information on how firms should mitigate the risks associated with foreign exchange volatility. During the period of high foreign exchange fluctuations, relevant government agencies should engage in the corporate education of managers of firms regarding how to adopt hedging so as to be able to survive over time. Financial learning for passive firms is very important, while simplicity of policy interventions would be effective, as complex models may not be helpful to small, less structured firms.

## 8. Study Contribution and Avenue for Further Research

First, a major contribution of this study was the use of the Hayes' mediation model. The most used mediation model had been the Baron and Kenny's, but as we stated in the study, this method is rife with criticisms. This study adopts a method that takes into consideration the gaps of Barron and Kenny's approach. Second, surveys of firm behavior with regards to economic or financial risks are rare. We approached our study-case in such a way as to provide policymakers with a not-so-complex report, which is what policy requires. While we appreciate and rely on the complex mathematical computation on hedging, we provided a much more policy-friendly perspective, without compromising on scientific rigor. Third, we also feel that the attempt to develop usable scales for measuring firms' abilities to manage risks from foreign exchange fluctuations is novel. All question items developed were crafted in an original manner, taking cues from theories and similar empirical studies. Fourth, the variables selected and combined in the research model are not known to have been tested in a systematic manner. While the scales of foreign exchange management are new, firm age, size and survival have been used in various models, but have not been seen to serve as prime predictors for managing foreign exchange risks.

Having made our modest contribution, we suggest possible avenues for further research. There are two areas of importance with regards to this study. The first hinges on the fact that a lot of studies have been carried out on hedging. While this has brought about an abundance of literature in this subject area, a closer look on the literature shows that there is few-to-no consensus on what firms should do to mitigate against shocks brought about by fluctuations in the foreign exchange market, especially in Developing Nations. It is important that a systematic review or meta-analysis be carried out with emphasis methodology, policy, and managerial implications to the firm. On the second aspect, it is important that more studies focus on smaller firms and on firms in developing economies. These sorts of firms are less structured, less formal, and probably less aware, making them the most affected whenever there are fluctuations in the foreign exchange market. "Size of firms" being used as a moderator in a cross-country study would be excellent.

**Author Contributions:** Conceptualization, H.O. and C.E.; Data curation, H.O., A.I. and B.I.; Formal analysis, H.O., A.I. and B.I.; Investigation, H.O., C.E., A.I. and B.I.; Methodology, H.O., A.I. and B.I.; Project administration, C.E., A.I. and B.I.; Resources, C.E., A.I. and B.I.; Software, H.O. and A.I.; Validation, H.O.; Visualization, C.E., A.I. and B.I.; Writing—original draft, H.O., A.I. and B.I.; Writing—review & editing, H.O., C.E., A.I. and B.I.

**Funding:** This research received no external funding.

**Acknowledgments:** We acknowledge three anonymous reviewers and the editorial team for their depth and speed. We also do acknowledge Messrs Onwe Chukwuemeka, Edigbo Anthony and Egede Churchill for their mobilization and support during the field work.

**Conflicts of Interest:** The authors declare no conflict of interest.

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
