# Peer review of "Firm Size and Age mediating the Firm Survival-Hedging Effect: Hayes’ 3-Way Parallel Approach"

_sustainability, doi:10.3390/su11030887_

Round 1

Reviewer 1 Report

This research is about the effects of hedging activities on the firm survival. Using seminal data of hedging activities of firms (Transaction Risk Management, Translation Risk Management, Economic Risk Management) the authors show the evidence that hedging activities directly affect the firm survival in positive way. However, the methodology they employ and the way they describe  the methodology and data have siginificant room to improve and I want to makes some comment on them.

Firstly, the causality problem is not solved here yet. There are still endogeneity problems in the research design. There should be some other relations between the predictors and error terms of mediators. Without controlling the possibilities we cannot be 100 percent sure that the predictors causes mediators and thus generate the outcome. For example, some economic charateristics such as factors which makes economy of scale much important in survival and firm size becomes big and as a result big organization are good in the form for Tranksaction Risk management. This kind of possibilities are not controlled in the research model and this problem should be fixed.

Secondly, the authors should explain in more detail and describe about how they build the measures for hedging activities such as TRC, TRSL, and ECO. Also, fSurv shoulde explained as well. Without it the readers cannot fully understand the description in the Table 2.

Thirdly, in the line 245 on page 10 the authors mention figure 3 but the figure doesn't exist in main body.

Fourthly, I recommend to add separate tables for direct, indirec, and total effect. Table 3 is very difficult to read and need to be improved for better readability in intuitive way.

Author Response

Response to Reviewer 1 Comments

For your comments, we are very grateful. We must also state that your views on issues of methodology are very well appreciated and have been appropriately addressed. Your introductory paragraph shows proper grasp of the hypothetical issues set to be achieved.

Point 1: “Firstly, the causality problem is not solved here yet. There are still endogeneity problems in the research design. There should be some other relations between the predictors and error terms of mediators. Without controlling the possibilities we cannot be 100 percent sure that the predictors causes mediators and thus generate the outcome. For example, some economic charateristics such as factors which makes economy of scale much important in survival and firm size becomes big and as a result big organization are good in the form for Tranksaction Risk management. This kind of possibilities are not controlled in the research model and this problem should be fixed.”

Response 1: We wish to state that your views on this issues are very well accepted and remain foundational for causality research[1], [2][3]. While we agree that the presence of endogeneity defeats the establishment of causality, we beg to humbly disagree that our design has endogeneity problems. First we wish to state that there is actually a rarity in the use of survey designs in handling issues of firm behaviour. Nevertheless, we argue in the body of the manuscript that our adherence to this design is to account for measurement precision amongst firms, focusing on their behaviour. Second, while one may argue that other covariates may exist in predicting the moderators (missing variables as a cause of endogeneity), in this study of ours, we do not see such as supported by theory. This does not imply that we can categorically state that it is only managing risks that predicts the firm’s age and size, but if there are other predictors, we do not find them in theory. For instance, the Schumpeterian Growth Theory or Theory of Creative Destruction has that salient notion of “Innovate or Die”[4], which implies that managing risks proactively, which we defined as “innovation” or “(financial) learning” is a precursor to whether a firm grows or not. We operationalize all the forms of managing foreign exchange rate risks as a form of innovation or financial learning and could not theoretically include other covariates that could lead to the growth or demise of the firm. This meant that those variables which we could have used as controls were not captured in the research instrument.

More, as we have opined earlier, we understand that the issue of endogeneity can be very subjective and intuitive[5]–[7], yet should not be ignored if captured by theory [8], [9]. For instance, Harman’s test is used to check whether a variable is endogenous, but whether it shows that or not, does not mean that that is the case. We make the objectivity claim based on example from our own study. That is, in the second step flow of firm’s age as a predictor of survival, the Harman’s test could show an absence of endogeneity in the age-survival effect, while that may be true, based on the statistical test result, one cannot neglect the fact (based on theory) that firm’s size also plays a covariate role in the effect of firm’s age and firm survival. So, it could be expected that managing risk (Innovation or learning) should have controls for predicting the four outcome variables of this study, we do not find that in theory, but rather agree that the multiple mediators we used serve as perfect control/ covariates in predicting firm survival.

·         We have included some clarity in the body of the manuscript to make up for the thought that our model could contain “noise”. We have done so using the Harman Single Factor[10] test and the Common Latent Factor approaches[11]. (See appendices)

Point 2:Secondly, the authors should explain in more detail and describe about how they build the measures for hedging activities such as TRC, TRSL, and ECO. Also, fSurv shoulde explained as well. Without it the readers cannot fully understand the description in the Table 2.”

Response 2: We do appreciate your concerns and have explained in more details and descried how the measures of TRC, TRSL, ECO and fSurv were developed as we have gone to make Table 2 clearer.

Point 3: “Thirdly, in the line 245 on page 10 the authors mention figure 3 but the figure doesn't exist in main body.”

Response 3: On the mention of figure 3, we initially had the figure in the body of the work, but later removed it, rather opting to use it as a graphical abstract as suggested by the journal. Since we do not find any restriction on the use of diagram both as an illustration and as graphical abstract, we have included the diagram to give a clearer presentation of our results and to maintain the internal flow of our findings.

Point 4: “Fourthly, I recommend to add separate tables for direct, indirec, and total effect. Table 3 is very difficult to read and need to be improved for better readability in intuitive way.”

Response 4: On the difficulty of reading Table 3, we apologize. The table has been properly presented in landscape form as the characters have been enlarged to aid proper visibility. Thank you for spotting that out, it would definitely make it easier to read, thank you. On the separation of tables, we adopted this style to be parsimonious, as the template from the adopted approach would even appear repetitive

For example, we would require three of this, with three mediators, which we have combined as one:

** The test for Common Method Variance are attached.

Appendices

Appendix A: Harman’s Single Factor Test

Total Variance Explained

Factor

Initial Eigenvalues

Extraction Sums of Squared Loadings

Total

% of Variance

Cumulative %

Total

% of Variance

Cumulative %

1

3.990

26.603

26.603

3.344

22.296

22.296

2

3.465

23.101

49.704

3

2.520

16.803

66.506

4

1.750

11.665

78.171

5

.917

6.112

84.283

6

.695

4.634

88.917

7

.559

3.725

92.642

8

.300

2.001

94.643

9

.210

1.401

96.044

10

.180

1.199

97.243

11

.155

1.036

98.279

12

.093

.619

98.897

13

.066

.441

99.338

14

.061

.405

99.743

15

.039

.257

100.000

Extraction Method: Principal Axis   Factoring.

Factor Matrixa

Factor

1

TRC1

.637

TRC2

.731

TRC3

.770

TRC4

.504

TRSL1

.098

TRSL2

-.027

TRSL3

-.031

TRSL4

-.005

ECO1

.494

ECO2

.438

ECO3

.447

ECO4

.546

fSurv1

.555

fSurv2

.437

fSurv3

.338

Extraction Method: Principal Axis   Factoring.

a. 1 factors extracted. 10 iterations   required.

Appendix B: Common Latent Factor Output

Reviewer 2 Report

1. Introduction:

There is no research question stated in Introduction. Also the structure of the paper is not introduced in the last paragraph of Introduction. 

2. Theory review:

I think this section is too short and should be extended.

3. Methods

This section should be revised. The dependent, independent and control variables should be stated. The questionnaire items measuring each particular variable and the sources of the scales shall be stated in the text.

It is not clear how many questionnaires where distributed and the number of respondents who submitted the responses? A number of usable responses? Bias? 

4. Conclusions can be revised.

What is the novel contribution of the study? implications for practitioners and policy-makers? Avenues for further research.

I hope the comments will hlp to improve the paper.

Author Response

Response to Reviewer 2 Comments

For your comments, we are very grateful. We appreciate your views on the structure of the paper. Again, thank you for your comments, they have definitely improved the work.

Point 1: “Introduction: There is no research question stated in Introduction. Also the structure of the paper is not introduced in the last paragraph of Introduction.”

Response 1: We have added research questions in the introduction, and have also concluded the introduction with a summary of the paper structure.

Point 2:Theory review: I think this section is too short and should be extended.”

Response 2: We humbly state that this paper was anchored on multiples theories and concepts, and for parsimony, we had to use a means of infusing the concepts and theories from the very start of the paper, in line with best practice[1] (See Geletkanycz & Tepper, 2012). Albeit, we are grateful for your take. Thank you.

Point 3: “Methods: This section should be revised. The dependent, independent and control variables should be stated. The questionnaire items measuring each particular variable and the sources of the scales shall be stated in the text. It is not clear how many questionnaires where distributed and the number of respondents who submitted the responses? A number of usable responses? Bias?”

Response 3: The methods have been appropriately revised. We have clearly stated the latent and observed variables, as could be seen in the “Exploratory factor analyses” table. For parsimony, the question items measuring each construct is well stated in the table earlier referred to. Same with the style of having adopted the use of theoretical foundations. We have explained in clearer terms our data administration details and management. We assure you that there are no issues of bias, both in design or analyses. We covered random manufacturing within 13 sub-sectors, cutting across five states along diverse geographical space.

Point 4:Conclusions can be revised. What is the novel contribution of the study? Implications for practitioners and policy-makers? Avenues for further research.”

Response 4: We have included a new section on the contribution of our study and avenues for further studies, while a chunk of the conclusion have been removed so as to still keep the work compact.

We are very grateful for your comments.

Thank you.

Reviewer 3 Report

In generally it is excellent article and very good original and interesting considerations, which is consistent with the pattern of research. 

The weakness of this article is only some small text editing problems, especially quality of charts and table (for example: figure number 2 and table numer 3) and one standard should be used for paragraphs.   

The suggestion is only improve the organization of work, especially take into account the paragraphs and in a better form to present the formulas, figures and tables.

Overall evaluation: article after these a little minor changes it is suitable for publication.

Author Response

Response to Reviewer 3 Comments

For your comments, we are very grateful.

Point 1: “In generally it is excellent article and very good original and interesting considerations, which is consistent with the pattern of research.”

Response 1: Thank you so much for your comments on this paper, we do not take it lightly.

Point 2:The weakness of this article is only some small text editing problems, especially quality of charts and table (for example: figure number 2 and table numer 3) and one standard should be used for paragraphs..”

Response 2: We have addressed the concern on the clarity of our tables and figures, and have also adopted a uniform method of our paragraphs.

We have also made corrections on spelling errors as advised.

Point 3: “The suggestion is only improve the organization of work, especially take into account the paragraphs and in a better form to present the formulas, figures and tables.”

Response 3: We are very grateful for your views on our paper. We feel encouraged to do more, and we also do hope that good thought would be requited in a very friendly and helpful form.

Thank you so much.

Round 2

Reviewer 1 Report

Thanks for sincere response and revision.

Even though the endogeneity problem still exist, the explanation authors give can cover the least rationale for current use of methodology and convince enough to make the logic sense.

The addition of the explanation on variable development and the improvement for readability of Table 3 are greatly helpful for readers.

Also the change of Table 1 helps a lot.

thanks for the opportunity to read interesting research.

Reviewer 2 Report

ok